# Effect of Chitosan-Diosgenin Combination on Wound Healing

**DOI:** 10.3390/ijms24055049

**Published:** 2023-03-06

**Authors:** Lubomir Petrov, Olya Stoilova, Georgi Pramatarov, Hristiyana Kanzova, Elina Tsvetanova, Madlena Andreeva, Almira Georgieva, Dimitrinka Atanasova, Stanislav Philipov, Albena Alexandrova

**Affiliations:** 1National Sports Academy “Vassil Levski”, 21 Akad. Stefan Mladenov St., Studentski Grad, 1700 Sofia, Bulgaria; 2Laboratory of Bioactive Polymers, Institute of Polymers, Bulgarian Academy of Sciences, Akad. G. Bonchev St., bl. 103A, 1113 Sofia, Bulgaria; 3Laboratory of Free Radical Processes, Institute of Neurobiology, Bulgarian Academy of Sciences, 23 Akad. G. Bonchev St., 1113 Sofia, Bulgaria; 4Department of Synaptic Signaling and Communication, Institute of Neurobiology, Bulgarian Academy of Sciences, 23 Akad. G. Bonchev St., 1113 Sofia, Bulgaria; 5Department of Anatomy, Faculty of Medicine, Trakia University, 11 Armejska St., 6000 Stara Zagora, Bulgaria; 6Department of Anatomy, Histology, General and Clinical Pathology and Forensic Medicine, Faculty of Medicine, Sofia University St. Kl. Ohridski, Lozenets University Hospital, 1 Koziak St., 1000 Sofia, Bulgaria

**Keywords:** chitosan, diosgenin, murine excisional wound model, oxidative stress

## Abstract

The difficult-to-heal wounds continue to be a problem for modern medicine. Chitosan and diosgenin possess anti-inflammatory and antioxidant effects making them relevant substances for wound treatment. That is why this work aimed to study the effect of the combined application of chitosan and diosgenin on a mouse skin wound model. For the purpose, wounds (6 mm diameter) were made on mice’s backs and were treated for 9 days with one of the following: 50% ethanol (control), polyethylene glycol (PEG) in 50% ethanol, chitosan and PEG in 50% ethanol (Chs), diosgenin and PEG in 50% ethanol (Dg) and chitosan, diosgenin and PEG in 50% ethanol (ChsDg). Before the first treatment and on the 3rd, 6th and 9th days, the wounds were photographed and their area was determined. On the 9th day, animals were euthanized and wounds’ tissues were excised for histological analysis. In addition, the lipid peroxidation (LPO), protein oxidation (POx) and total glutathione (tGSH) levels were measured. The results showed that ChsDg had the most pronounced overall effect on wound area reduction, followed by Chs and PEG. Moreover, the application of ChsDg maintained high levels of tGSH in wound tissues, compared to other substances. It was shown that all tested substances, except ethanol, reduced POx comparable to intact skin levels. Therefore, the combined application of chitosan and diosgenin is a very promising and effective medication for wound healing.

## 1. Introduction

Skin injuries are referred to as wounds [1]. Both acute and chronic wounds fall under this category. A sudden injury to the skin is known as an acute wound. It can be cured in two to three months depending on its size and depth in the epidermis or dermis layers of the skin [2]. In addition, burns, leg ulcers, decubitus ulcers, infections, etc., are life-threatening chronic wounds because they do not heal quickly [1]. Accordingly, the difficult-to-heal wounds continue to be a serious problem for modern medicine. Therefore, the development of more advanced, affordable and cost-effective wound dressings is urgently needed.

The healing process begins as soon as a wound develops. The phases that overlap and interact during the complex, dynamic and wound repair process are coagulation, immunological response, inflammation, proliferation and remodeling [3,4]. The inflammatory phase is characterized by the migration of inflammatory cells to the wound to defend against pathogens followed by activation of the skin cells. Neutrophils and macrophages are stimulated during this phase by producing pro-inflammatory cytokines, including interleukin-1β, tumor necrosis factor, as well platelet-derived, transforming and fibroblast growth factors [5,6,7]. The wound-healing process involves the participation of reactive oxygen species (ROS). They are powerful oxidizing agents and significant sources of cell deterioration, but also serve positive functions and are essential for setting up the normal wound healing response [8,9,10]. By generating cell-survival signaling, low levels of ROS help to protect tissues from infection and promote efficient wound healing [11,12]. However, excessive ROS generation can exceed the ability of endogenous antioxidants to scavenge them causing oxidative stress (OS). This redox imbalance damages cells, promotes inflammation and dysregulates the healing process [13]. Consequently, a good equilibrium between low and high ROS levels is crucial for the wound-healing process.

It is commonly accepted that the ideal wound covering should mimic many properties of human skin—adhesion, elasticity, durability and impermeability to bacteria [14]. Hence, bioactive polymers have received considerable attention as effective wound-healing agents not only because of their biocompatibility and biodegradability, but also because they have an active therapeutic effect on one or more stages of wound healing [15]. One of the most studied and promising biopolymers for wound healing purposes is chitosan [16]. Chitosan, a renewable natural polysaccharide, composed of randomly distributed N-acetyl-D-glucosamine (*N*-acetylglucosamine) and 2-amino-2-deoxy-glucose (glucosamine) units, represents excellent biocompatibility, biodegradability, hemostatic, mucoadhesive and wound healing properties, as well anti-inflammatory, antioxidant, antimicrobial activity [17]. In addition, it is believed that chitosan hastens the production of fibroblasts and aids in the initial stages of healing [18,19]. Due to these benefits, and the ability to absorb exudates and film forming properties, chitosan is a good candidate in wound healing applications.

Although it has been demonstrated that chitosan encourages tissue growth and differentiation during wound healing, its usage in wound care is impeded due to its poor mechanical properties [20,21,22]. In order to overcome this drawback, numerous synthetic polymers are used. Among them, polyethylene glycols (PEG) are widely used as plasticizers. Due to their advantageous properties, such as biocompatibility, solubility and low-toxicity, PEGs are suitable for contact with living organisms. In addition, their incorporation improves the release of poorly water-soluble bioactive substances and improves the therapeutic efficacy of various medications [23].

Diosgenin (25R-spirost-5-en-3-ol) (Figure 1), a hydrolysate of dioscin, is a naturally occurring steroidal saponin that is present in a number of plants, including *Trigonella foenum graecum*, *Solanum incanum*, *Solanum xanthocarpum*, *Smilax china Linn* and *Dioscorea nipponoca* Makino [24,25]. As a physiologically active phytochemical, diosgenin (Dg) has a variety of effects on plants [26]. It is utilized as a medicine to treat conditions such as diabetes, hypercholesterolemia, climacteric syndrome, leukemia and cancer, as well in the production of steroids [27]. Dg possesses high biocompatibility, immune-protection, anti-inflammatory and antioxidant properties [28,29,30]. Compared to the conventional antioxidant activity of vitamin C, diosgenin extracted from *Costus speciosus* has an efficient antioxidant scavenging affinity against DPPH radicals [31]. Diosgenin also enhances the antioxidant status, prevents lipid peroxidation and reduces inflammation by preventing the generation of cytokines, enzymes, and adhesion molecules which promote inflammation [32]. However, the pharmaceutical applications of diosgenin are very limited because of its poor pharmacokinetics and extremely poor water solubility [33,34]. Recently, it was shown that encapsulation of Dg in chitosan/bovine serum albumin nanoparticles enhanced its bioavailability [34]. Therefore, finding a suitable combination of diosgenin and chitosan will be an alternative strategy to overcome the limitations of diosgenin applications and to improve the healing effect of both substances.

The testing of various combinations of substances that accelerate the healing of wounds is the basis for creating new more effective therapies. For this reason, in the present study, it is aimed to investigate the wound healing effect of the combined administration of chitosan and diosgenin on a mouse skin wound model. To the best of our knowledge, there are no publication on the simultaneous application of chitosan and diosgenin in the treatment of wounds. Given the role of inflammation and oxidative stress in wound regeneration, as well the anti-inflammatory and antioxidant effects of chitosan and diosgenin, the lipid peroxidation (LPO), protein oxidation (POx) and total glutathione (tGSH) levels were evaluated. In that way, it was shown that the proposed combination of chitosan and diosgenin, is a very promising formulation with significant potential towards wound dressing and healing applications.

## 2. Results

### 2.1. Macroscopic Observations

The wound healing effect of the chitosan-diosgenin combination was evaluated by treatment of the wounds made on the mice dorsum. The macroscopic analysis was performed by measuring the wound areas photographed on the 3rd (D3), 6th (D6), and 9th (D9) days and compared with those before the first treatment (D1). Figure 2 shows the macroscopic appearance of wounds treated respectively with polyethylene glycol in 50% ethanol (PEG), chitosan and PEG in 50% ethanol (Chs), diosgenin and PEG in 50% ethanol (Dg) and chitosan, diosgenin and PEG in 50% ethanol (ChsDg) at D1 and D9. In this experiment, a 50% ethanol solution was used as a control. Complete schematic presentation of wound treatment is explained and shown in Section 4.3. All mice survived throughout the period until sacrifice. There was no evidence of necrosis. Clearly, on the 9th day the majority of the wounds appeared to be healed and were reduced significantly in size without external signs of inflammation (Figure 2). The macroscopic appearance of the wounds during the D3 and D6 treatment can be observed in Appendix A.

Furthermore, the relative reduction of wound areas vs. the initial wound areas (on D1) after treatment with various combinations is illustrated in Figure 3. The macroscopic results indicated faster healing in the animals tested with Chs and ChsDg. In particular, Chs and ChsDg combinations gave a significantly greater reduction in wound area as early as day 3 (67.2 and 62.6%, respectively) compared to PEG (32.4%) and Dg (8.1%). On the 9th day the largest decrease of the wound area was observed in the group treated with ChsDg (82%). The difference in the treated wound areas with Chs and ChsDg was apparent on the 9th day. The advantage is that the measured mean relative contraction of wounds treated with ChsDg was bigger than this treated with Chs (59.2%). PEG also had a good effect, similar to this of ChsDg, but it appeared with a delay—on the 6th day. Thus, in overall, the ChsDg combination demonstrated a better wound healing effect.

### 2.2. Microscopic Observations

The changes on the desquamated epithelium regions, erythema, rhagades, loss of skin appendages and corrosive skin effect are listed in Table 1 and Table 2. In addition, the representative histological photomicrographs of the structural organization of test groups are shown in Figure 4. The application of ethanol was accompanied by moderate erythema, a weak skin corrosive effect and the presence of desquamation. Compared with the control, the application of PEG, Dg and Chs gave mild erythema and desquamation. In addition, the treatment of skin wounds with PEG and Dg leads to mild skin corrosive effect, while treatment with PEG and Chs leads to rhagades and focal loss of skin appendages.

Healing tests performed in mice using various combinations clearly showed that the ChsDg combination may contribute to faster wound healing without complications.

### 2.3. Oxidative Stress

Regarding the indicators of oxidative stress, no differences were detected in LPO levels between intact skin and control skin. LPO was significantly lower in PEG-treated wounds and higher in Chs- and ChsDg-treated wounds compared to tissue from ethanol-treated (Control) wounds (Figure 5). Of the examined substances, only the administration of Dg retained the levels of LPO similar to control.

The POx was elevated in ethanol-treated wounds (Control) compared to levels in intact skin. POx values close to those in intact skin were measured in the wounds treated with the tested substances (Figure 6).

The concentration of tGSH was non-significantly higher in ethanol-treated wounds (Control) compared to untreated skin (Intact). The application of the investigated substances resulted in a decrease in the concentration of tGSH in wounds except for the ChsDg combination where no significant difference was reported compared to the intact skin samples and the wounds treated with ethanol (Figure 7).

## 3. Discussion

Wound healing is known to be a complex and dynamic process, regulated by various physiological and biochemical parameters, which act together to promote tissue restoration [15]. This process can happen more quickly and efficiently with appropriate formulations. In this context, topical formulations based on chitosan attract much attention because of their known active role in the first three stages of wound healing—hemostasis, inflammation, proliferation [19,35]. Moreover, chitosan-based hydrogels may aid the re-establishment of skin architecture and its degradation by-products are non-cytotoxic [36]. It is observed that regardless of the recognized healing effect of chitosan, the number of studies on its various combinations with biological agents is increasing, with the aim of improving this effect. Despite the wide variety of chitosan formulations investigated, the synergistic effect of chitosan and diosgenin on wound healing has not yet been reported.

In this study, the focus was on the preparation of a chitosan-diosgenin combination and studying its effect on the wound healing process. For this reason, various combinations were tested—polyethylene glycol in 50% ethanol (PEG), chitosan and PEG in 50% ethanol (Chs), diosgenin and PEG in 50% ethanol (Dg) and chitosan, diosgenin and PEG in 50% ethanol (ChsDg), respectively. The optimum chitosan concentration (3% *w*/*v*) and water to ethanol ratio (1/1 *v*/*v*) required for film formation were found by varying the ratios. Moreover, lactic acid was used in chitosan dissolution due to its plasticizer characteristic, which gives lower stiffness and a higher percentage of elongation, besides helping the antimicrobial properties. In order to prepare ChsDg combination, the poorly water-soluble diosgenin was firstly dissolved in ethanol, then mixed with the PEG solution in ethanol and finally this solution was added to the aqueous chitosan solution. The use of 50% ethanol has an additional advantage because it minimizes the drying time and promotes faster film formation. The prepared colorless viscous chitosan solution was characterized by dynamic viscosity of 1780 cP and storage module (G’) lower than loss module (G”). Therefore, this chitosan solution was a physical gel with low mechanical properties, which is in accordance with the literature [37]. Hence, in all tested combinations an equal amount of PEG was used as plasticizer.

Various wound models have been developed, considering the obstacles in sampling acute skin wounds in humans due to ethical considerations. The use of organotypic-cultured skin approximates the natural process, but does not reflect the overall response of the organism and is influenced by external factors such as the composition of the culture medium [38]. Animal models are the available alternative to study the complex interactions at molecular and cellular levels that occur during the wound healing process in a biologically relevant environment [39]. However, differences in thickness and composition between animal and human skin should be considered when interpreting the obtained data. Indeed, there is no animal model that represents all aspects of wound healing seen in humans. It is assumed that the main disadvantage of the excisional wound model in rodents is that the healing process is through the contraction of the panniculus carnosus while the human wound heals through re-epithelization. However, a disadvantage of the splint model is the possibility of self-mutilation by the animals through the ring, while the protective dressings are difficult to fix due to the movement and activity of the animals. For that reason, in this study the excisional wound model without silicon splint was applied, in accordance to the similar models in the literature [40,41,42]. In addition, some authors revised the considered limits of the excisional wound model because of the perception that rodent wounds heal by contraction while humans heal by re-epithelialization [43]. The data have shown that contraction occurs only after epithelial closure and the notion of the domination of closure by contraction is rather inaccurate. Thus, simple murine excisional wounds provide a valid and reproducible model that heals by both contraction and re-epithelialization [43].

The macroscopic results indicated that the greatest effect on wound retraction was from Chs and ChsDg, followed by PEG. Treatment with Chs and ChsDg led to faster retraction of the wounds—already on the 3rd day, while the effect of PEG reached that of Chs and ChsDg on the 6th day, and the overall effect on the 9th day of ChsDg was larger. These results were expected, because chitosan-based gel formulations have proven antioxidant and anti-inflammatory activities [19,35,36]. It is noteworthy that during the wound healing process, chitosan gradually depolymerizes to release *N*-acetylglucosamine which initiates fibroblast proliferation [36]. In this way collagen deposition is arranged, stimulating the synthesis of natural hyaluronic acid at high levels at the wound site [36]. Likewise, Dg with its immune-protection and antioxidant status, prevents lipid peroxidation and reduces inflammation [27,29]. In addition, a similar retracting effect of PEG, Chs and their combination was also found by other authors in the therapy of experimental wounds [44]. The histological criteria give an idea of the damage to the skin around the wounds by the studied combinations, which is decisive for their applicability in practice. Moreover, it was observed that chitosan-based gel formulation containing diosgenin was more effective compared to the other-treated groups, especially on the 9th day (Figure 2). Similarly, when wound area shrinkage was examined, recovery in the case of ChsDg was faster (Figure 3). The observed minimal irritant effect of the combination ChsDg (Table 1 and Table 2) and the good therapeutic effects on the experimental wounds (Figure 3) give grounds for continuing studies on the possibilities of using this combination in practice. Apparently, the chitosan-diosgenin combination accelerates wound healing by exhibiting synergistic activity.

The implication of chitosan in wound healing is related to its interaction properties with neutrophils, resulting in IL-8 secretion that causes the migration of neutrophils to chitosan [45]. The immunomodulatory effects of chitosan are relatively broad, as it can induce a plethora of cytokines of a pro- or anti-inflammatory nature. Whether these responses are negative or positive depends on various factors. The infiltration of inflammatory cells is essential for wound repair, but excessive cell infiltration accompanied by classical immune cell activation can result in tissue damage [46]. There are data that activated neutrophils generate O_2_^•−^ which induces lipid peroxidation and stimulates collagen synthesis [47]. This may explain the weak but significantly elevated LPO levels in the wounds treated with Chs and ChsDg. Given the observed beneficial effect of Chs and ChsDg in the wound healing process, it is likely that these small amounts of ROS play a rather positive signaling role [11].

The total protein carbonyl content is one of the most important biomarkers providing a qualitative assessment of the presence of OS in vivo [48]. Elevated protein carbonyl levels measured from biological tissues and fluids are associated with decreased enzyme activities, loss of protein function, inflammation activity, etc. [49]. The high level of protein carbonyl groups in the control samples observed in this study indicate that oxidative processes have taken place to some extent. The application of the tested substances (Dg, Chs, ChsDg) led to levels of protein carbonyl groups like that of the intact skin. This finding implies a suppressed inflammatory activity by these components [32,50]. The anti-inflammatory action of Dg could be explained through the stimulation of IFN-γ production [51] and the antioxidant ability of Chs is affiliated with the protonated NH_2_-groups, which are responsible for free radical scavenging [52]. The wound-healing process was better when both were combined.

It should be mentioned that the application of Chs and the ChsDg on wounds leads to LPO increase but suppresses the POx (Figure 5 and Figure 6). Probably, the observed significant rise of MDA in contrast to protein carbonyls in Chs-treated wounds is due to the mechanism of LPO that occurs as a chain process after initiation by ROS and the high sensitivity of polyunsaturated fatty acids to oxidative damage [53]. Furthermore, proteostasis includes both protein stabilization and preferential degradation of the modified proteins [54]. Another explanation for the observed increase in MDA is a possible direct reaction of residual chitosan in the wound samples with the thiobarbituric acid used in the LPO assay method [55].

Glutathione is the most prevalent intracellular antioxidant and provides significant protection to the tissue from oxidative stress [56]. Glutathione can act directly as a ROS scavenger or indirectly through its participation as a co-substrate in antioxidant enzyme reactions. Particularly, GSH acts as an electron donor to hydrogen peroxide, forming an oxidized thiol, which is recycled by glutathione reductase. In this way, the antioxidant concentration is maintained at the wound site through regeneration [57]. Another potential reaction is the degradation and excretion from the cell of the oxidized thiol via glutathione-S-transferase [56]. In wounds, the GSH level is significantly lower when compared to undamaged skin [56]. Glutathione is essential for the wound-healing process. It was demonstrated that low antioxidant (incl. GSH) levels play a significant role in delaying wound healing in either aged or diabetic rats [58,59]. The obtained results indicate that administration of the chitosan-diosgenin combination resulted in the highest levels of GSH compared to the other substances tested (Figure 7). This further confirms that both chitosan and diosgenin accelerate wound healing by their synergistic activity. Moreover, the ChsDg combination might contribute wound healing by preventing the prolongation of the inflammation phase with their anti-inflammatory effects.

## 4. Materials and Methods

### 4.1. Materials

Commercially available chitosan (Chs, medium molecular weight, viscosity 200–400 cps, 90% degree of deacetylation) was supplied by Yantai Shang Tai Trading Co., Ltd, Yantai, China. Thiobarbituric acid, 5,5′-dithiobis-2-nitrobenzoic acid (DTNB), 2,4-dinitrophenylhydrazine (2,4-DNPH), diosgenin (Dg) and polyethylene glycol (PEG, average molecular weight 400 g/mol) were purchased from Sigma-Aldrich (St. Louis, MI, USA). All chemicals and solvents were of analytical grade and used without further purification.

### 4.2. Preparation of the Combinations

The chitosan in PEG (Chs) combination was prepared by dispersing chitosan powder (3% *w*/*v*) in distilled water followed by adding lactic acid under stirring until the complete chitosan dissolution. Subsequently, PEG solution in ethanol (0.16% *w*/*v*) was added to the chitosan solution. The total water:ethanol volume ratio was 1:1.The pH of the prepared solution was 4.

In order to prepare the chitosan and diosgenin in PEG combination (ChsDg), a solution of PEG in ethanol (0.16% *w*/*v*) and solution of diosgenin in ethanol (0.1% *w*/*v*) were added to the aqueous chitosan solution (3% *w*/*v*) in lactic acid. The total water:ethanol volume ratio in the prepared ChsDg combination was again 1:1.

Blank diosgenin solution in PEG (Dg) in 50% ethanol (0.1% *w*/*v*) and PEG in 50% ethanol (0.16% *w*/*v*) were also prepared. As well, a control of 50% ethanol was used.

### 4.3. In Vivo Experiments

In vivo experiments were conducted on 10 albino mice, 25–35 g (10 weeks of age). The mice were kept in a controlled temperature room (24 ± 2 °C) with a 12 h light/dark cycle and with free access to food and water. Before treatments mice were anesthetized with Xylazine (20 mg·kg^−1^) and Ketamine (100 mg·kg^−1^). The entire back of the experimental animals was depilated with a trimmer and subsequent total epilation using an epilating cosmetic cream. After depilation, four identical wounds were made with a 6 mm diameter punch. The wounds of 5 of the animals were treated on days 1, 3, and 6 either with 50% ethanol (Control), a solution of PEG in 50% ethanol (PEG), a solution of chitosan and PEG in 50% ethanol (Chs) and combined solution of chitosan, diosgenin and PEG in 50% ethanol (ChsDg). In the remaining 5 mice, one of the wounds was treated with a solution of diosgenin and PEG in 50% ethanol (Dg) instead of PEG (Figure 8). The four wounds were made on the dorsum of the animals and were treated with the various substances studied to make a more objective comparison of the effect of substances on the healing process, avoiding individual differences between animals. The wounds were treated for the first time immediately after the wounds were made. Then, before the next treatment, the wounds were cleaned with 50% ethanol and then re-applied with the same substances. This procedure was repeated at 3 and 6 (D3 and D6) days.

Before the first treatment (D1) and on the 3rd (D3), 6th (D6) and 9th (D9) days pictures of wounds were taken using a digital camera, and the wound areas (mm^2^) were determined using the image analysis program Image J. On day 9, animals were euthanized. An intact skin and wounds from 4 animals were excised for histological analysis and those from 6 animals for measurement of lipid peroxidation (LP), protein oxidation (POx) and total glutathione (tGSH) levels. The mice were maintained and used in accordance with the guidelines of the Care and Use of Laboratory Animals (US National Institute of Health) and the rules of the Ethics Committee of the Institute of Neurobiology, Bulgarian Academy of Sciences (registration FWA 00003059 by the US Department of Health and Human Services).

For histological purposes, tissue samples from the wound sites treated with the tested substances were fixed in 10% neutral buffered formalin overnight at 4 °C. Thereafter, the tissues were embedded in paraffin and cut into 6 µm thick sections. The samples were then deparaffinized with xylene and ethanol and processed for the classical histological staining hematoxylin and eosin (H&E). After the reaction, the sections were dehydrated in ethanol, cleared in xylene and coverslipped with Entellan (Merck, Darmstadt, Germany). The slides were observed and carefully photographed with a Nikon research microscope equipped with a digital camera DXM1200c.

Equal-weighted skin samples were carefully cut with scissors into small pieces and then homogenized with a D-160 homogenizer (DLAB Scientific Inc., City of Industry, CA, USA). After centrifugation at 3000 rpm, the post-nuclear tissue homogenate was used to measure the total protein content and the tested oxidative stress parameters: lipid peroxidation, protein oxidation, and total glutathione. The Biuret method based on a colorimetric test for total proteins was used to determine the protein content. The 1.10307 Protein Kit was purchased from Sigma-Aldrich (St. Louis, MI, USA). The lipid peroxidation assay was based on the reaction between the end-products of the LPO and thiobarbituric acid according to Ben Mansour (2011) with some modifications [60]. The post-nuclear homogenates of the skin (mg protein ml^−1^) in 0.15 M KCl-10 mM potassium phosphate buffer, pH 7.2, were heated for 15 min at 100 °C in the presence of 40% trichloroacetic acid, 5N HCl and 2% thiobarbituric acid (2:1:2 *v*/*v*) for color developing. The absorbance was measured at 532 nm against dd H_2_O after cooling and centrifugation. The values were expressed in nmoles malondialdehyde (MDA) per mg protein, using a molar extinction coefficient of 1.56 × 10^5^ M^−1^ cm^−1^.

Protein carbonyl (PC) groups were quantified by reaction with 2,4-dinitrophenylhydrazine (2,4-DNPH), according to the method of Whitekus et al. (2002) with some modifications applied [61]. About 0.2 mL of homogenate without (control) and with 1 mL of 10 mM 2,4-DNPH (in 2 M HCl) were incubated at 37 °C for 90 min. After adding 1 mL of 28% trichloroacetic acid, samples were vortexed for 1 min and then centrifuged for 10 min at 3000× *g*. The precipitates were washed with ethanol-ethyl acetate in a 1:1 ratio 3 times. Washed samples were then dissolved in 6 M guanidine (in 20 mM potassium phosphate buffer, pH 2.3, adjusted with HCl). Absorbance at 360 nm was measured, and carbonyl content was expressed as nmoles carbonyl.mg^−1^ protein using a molar extinction coefficient of 2.2 × 10^4^ M^−1^ cm^−1^.

The amount of tGSH was determined by the method described by Rahman et al. [62]. The absorption at 412 nm of the color compound 5′-thio-2-nitrobenzoic acid (TNB) resulting by the reaction between reduced glutathione (GSH) and 5,5′-dithiobis-2-nitrobenzoic acid (DTNB) was read. The reaction rate is proportional to the amount of the reduced glutathione in the sample. The amount of tGSH present was calculated using the standard and represented as ng/mg protein.

### 4.4. Statistical Analyses

Descriptive statistics, the Shapiro–Wilks test of normality and One-Way ANOVA with Tukey post hoc test were applied using the statistical program GraphPad Prism 7.0. In the text, all data are presented as the mean ± standard deviation (SD) and in the figures as the mean ± standard error of measurement (SEM).

## 5. Conclusions

For the first time, the combined action of chitosan and diosgenin on wound healing was studied. The obtained results indicate that the chitosan-diosgenin combination enhanced the regenerative effect and gave the largest and the most rapid reduction in wound area. Regarding oxidative indicators, the chitosan-diosgenin combination maintained high levels of the non-enzymatic antioxidant glutathione in wound tissues compared to the other substances tested. All tested substances reduced protein oxidation to values comparable to levels in intact skin. Furthermore, the diosgenin antioxidant effect successfully complemented chitosan’s good retracting effect in wound healing. In this way, it was shown that the proposed combination of chitosan and diosgenin is a very promising formulation with significant potential towards wound dressing and healing applications.

## Figures and Tables

**Figure 1 ijms-24-05049-f001:**
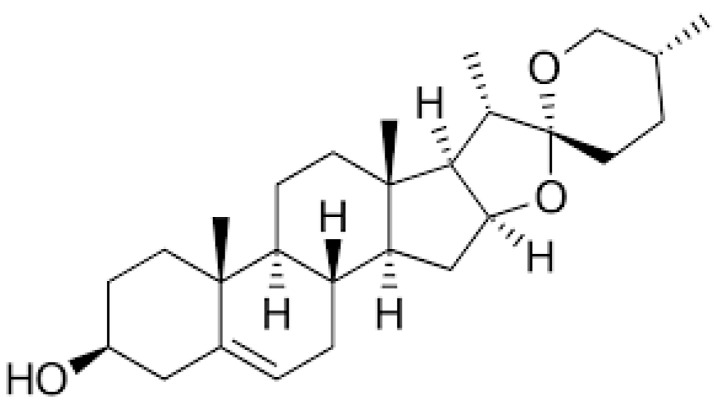
Chemical structure of diosgenin.

**Figure 2 ijms-24-05049-f002:**
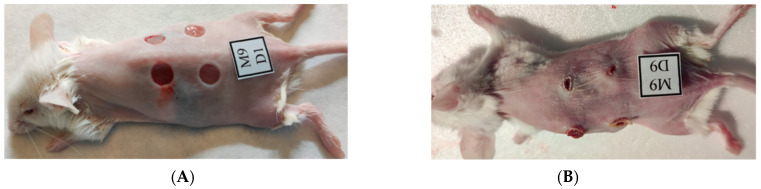
Digital imaging of the wound areas before (**A**) and after (**B**) treatment at D1 and D9, respectively.

**Figure 3 ijms-24-05049-f003:**
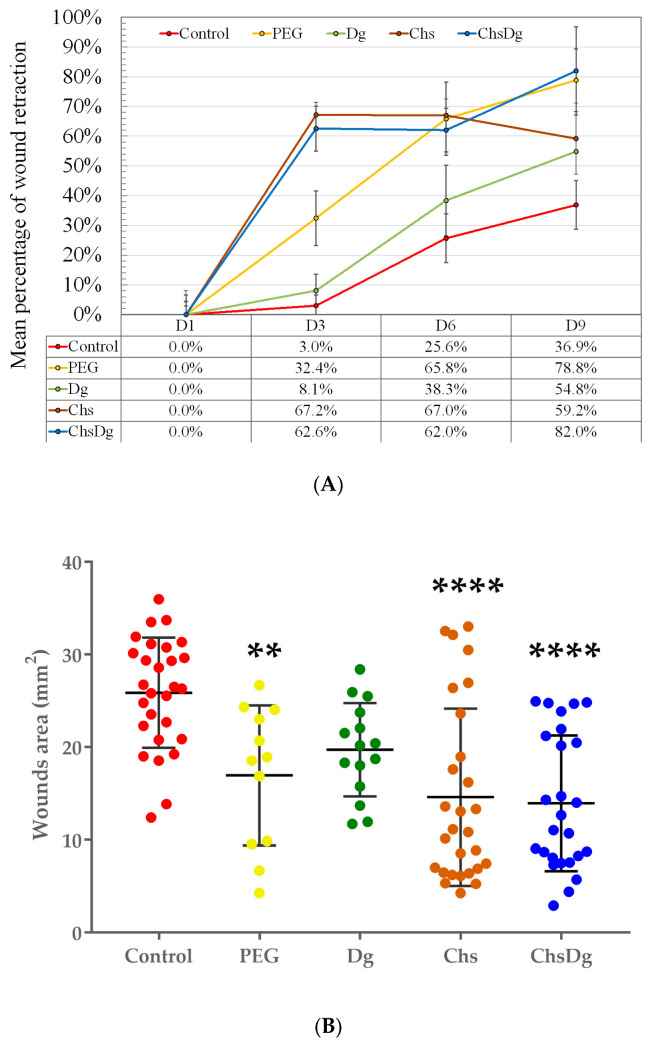
Dynamics of wound areas reduction during treatment with various combinations (**A**). ANOVA statistics for the significance of wounds’ area for the entire course of treatment (**B**). **—*p* < 0.01; ****—*p* < 0.0001 vs. Control.

**Figure 4 ijms-24-05049-f004:**
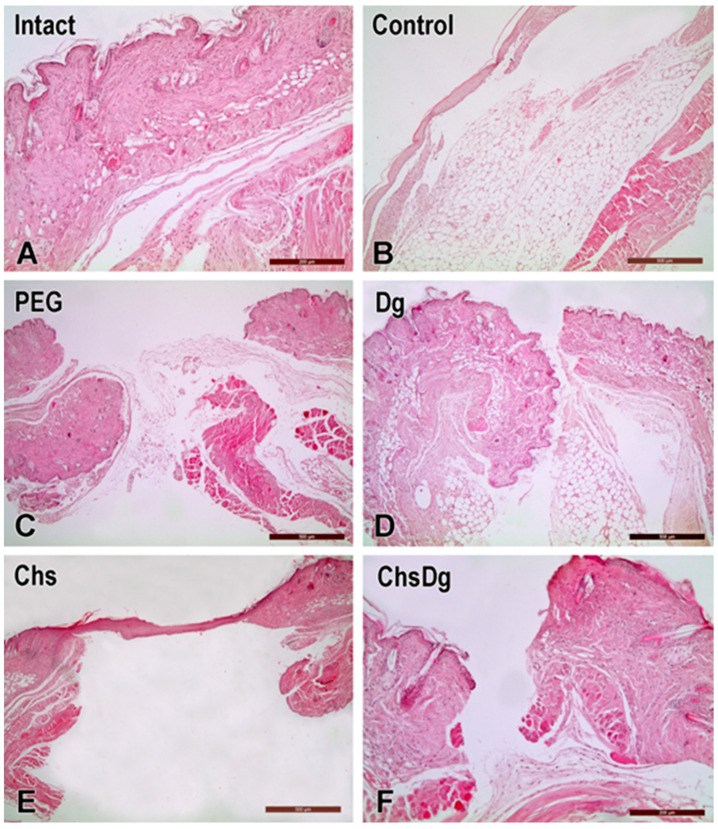
Representative hematoxylin and eosin stained photomicrographs of the structural organization of (**A**) mouse intact skin, (**B**) wound treated with 50% ethanol, which served as a control, (**C**) wound treated with PEG, (**D**) wound treated with Dg, (**E**) wound treated with Chs, (**F**) wound treated with ChsDg. Scale bars: 200 µm (**A**,**F**), 500 µm (**B**–**E**).

**Figure 5 ijms-24-05049-f005:**
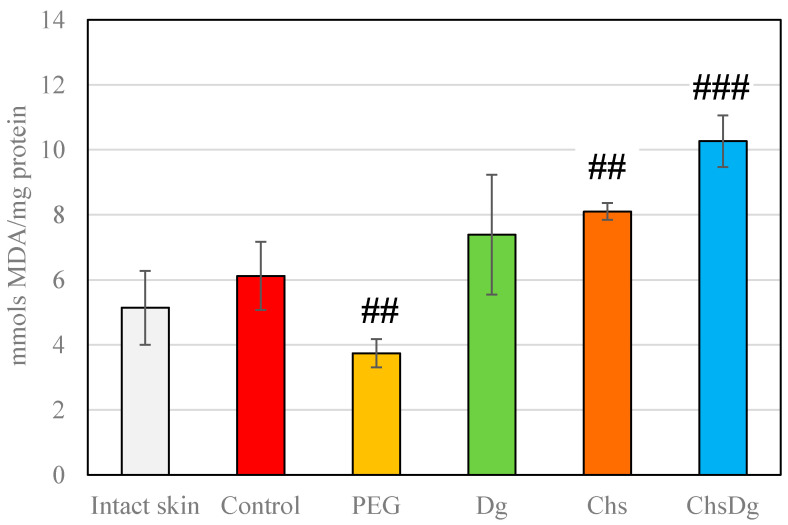
Lipid peroxidation in intact skin and wound tissues treated with the investigated substances; ##—*p* < 0.01, ###—*p* < 0.001 vs. control.

**Figure 6 ijms-24-05049-f006:**
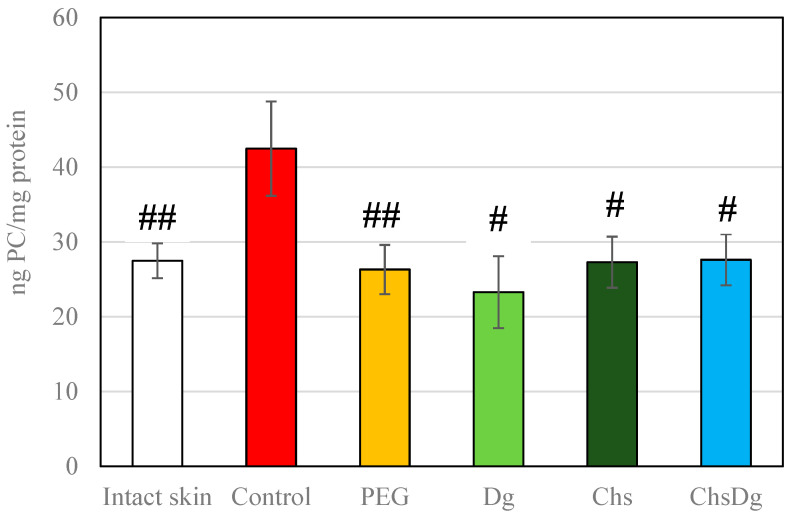
Protein oxidation in intact skin and wound tissues treated with the investigated substances; #—*p* < 0.05, ##—*p* < 0.01 vs. control.

**Figure 7 ijms-24-05049-f007:**
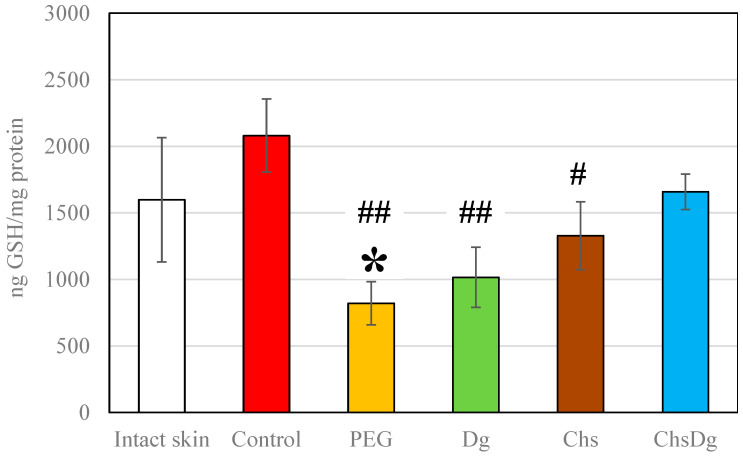
Concentration of total glutathione in intact skin and wound tissues treated with the investigated substances; #—*p* < 0.05, ##—*p* < 0.01 vs. control; *—*p* < 0.05 vs. intact skin.

**Figure 8 ijms-24-05049-f008:**
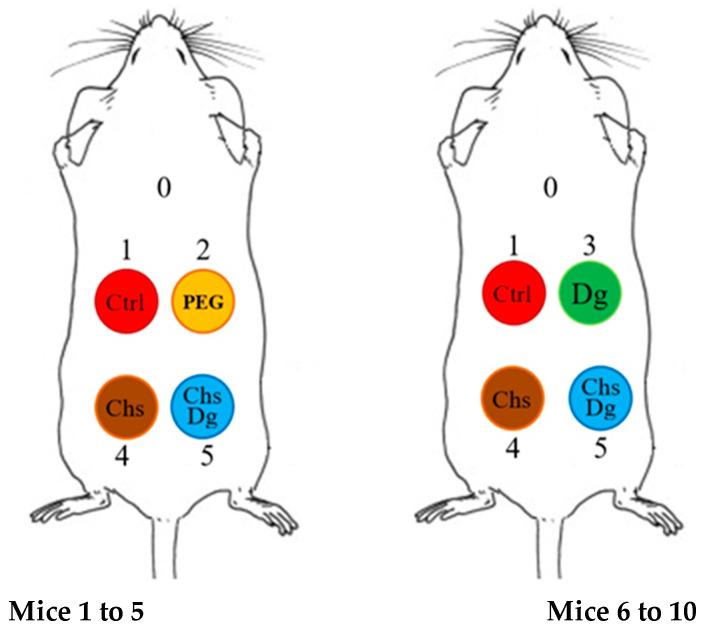
Scheme of the wounds treatment.

**Table 1 ijms-24-05049-t001:** Skin status regarding erythema and corrosive effect.

Parameters	Intact	Control	PEG	Dg	Chs	ChsDg
Erythema						
Absence	0					
Mild			1	1	1	1
Moderately expressed with slight edema		2				
Strongly expressed with edema						
Pronounced with swelling crossing the border of the treated field						
Skin corrosive effect/after repeated contact/						
Absence	0				0	0
Mild		1	1	1		
Moderate						
Intense/Strong						
Very strong						
Ratio of stratum corneum:epidermis:papillary layer of the dermis	1:10:20	1:5:10	1:8:15	1:8:15	1:8:15	1:5:10

**Table 2 ijms-24-05049-t002:** Skin status regarding desquamation, rhagades, and loss of skin appendages.

Parameters	Intact	Control	PEG	Dg	Chs	ChsDg
Desquamation	−	+	+	+	+	−
Rhagades	−	−	+	−	+	−
Loss of skin appendages	−	−	Focal	−	Focal	Focal

## Data Availability

The data presented in this study are available on request from the corresponding author.

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
