# Peer review of "Effect of Chitosan-Diosgenin Combination on Wound Healing"

_ijms, 2023, doi:10.3390/ijms24055049_

Round 1
Reviewer 1 Report
The author presented a comparative study of applications of 50% ethanol, polyethyleneglycol (PEG) in 50% 24 ethanol, chitosan in PEG (Chs), diosgenin in ethanol (Dg) and chitosan/diosgenin (ChsDg) in wound healing.
The main problem addressed in this research is wound healing using different agents, the better results are obtained for chitosan/diosgenin. The materials used novel, and the results are relevant in the field. Chitosan/diosgenin combination is unique, it has been rarely researched so far. However, this material has to be investigated in detail. Detailed investigation on chitosan/diosgenin is suggested, like scientific investigations, enzyme activities, immunological investigations, and preparing and testing wound dressing.
References on chitosan/diosgenin has to be cited and discussed. The authors have to present a comprehensive study on the use of chitosan/diosgenin in wound healing in addition to antibacterial and antifungal activities. Additional data sets and figures regarding chitosan/diosgenin are required.
The conclusion has to revise after adding new results and inputs from suggested studies.
Author Response
POINT-TO-POINT RESPONSE TO REVIEWER 1 COMMENTS
On behalf of all the co-authors I gratefully thank the Reviewer 1 for the thorough analysis of our manuscript, as well as for the comments. Hereafter our responses and the changes made in the manuscript according to the recommendations of the Reviewer 1 are reported. All modifications were mark with the Track Changes tool.
Response to Reviewer 1 Comments
1) References on chitosan/diosgenin has to be cited and discussed. The authors have to present a comprehensive study on the use of chitosan/diosgenin in wound healing in addition to antibacterial and antifungal activities.
Response 1: To the best of our knowledge, there is a lack of data in the literature on the effect of chitosan/diosgenin on wound healing. In this regard, in the present study we describe this effect. Moreover, based on the anti-inflammatory and antioxidant effects of diosgenin and chitosan separately, as well the involvement of oxidative stress in the inflammatory response of wounds, we have studied the effect of chitosan/diosgenin administration in terms of changes in the oxidative status of wound tissues.
2) Additional data sets and figures regarding chitosan/diosgenin are required.
Response 2: According to the referee comment, the ANOVA results have been included as Figure 3B in the revised manuscript, as well as the results explanation.
3) The conclusion has to revise after adding new results and inputs from suggested studies.
Response 3: As suggested by Referee 1 Conclusions have been rewritten with more details and comments on new results in the revised version.

Reviewer 2 Report
Authors have demonstrated the combination of chitosan and diosgenin in wound healing. Both components are well-known as wound-healing materials. Authors have used these materials directly on the animal model to show their skin regeneration potential. The manuscript can be published after minor revisions.
1) Why has PEG wound healing potential similar to ChsDG on day 9?
2) There is no significant difference in the wound area of Chs and ChsDg. What could be the advantages of using both?
3) Have authors observed immune cells in wound tissues?
4) Authors should show pro and anti-inflammatory macrophages in wound tissues through immuno-staining.
5) Keratin 14/5 expression in wound tissues should be shown as these are important for skin regeneration.
Author Response
POINT-TO-POINT RESPONSE TO REVIEWER 2 COMMENTS
On behalf of all the co-authors I gratefully thank the Reviewer 2 for the valuable comments and suggestions aimed at improving the quality of our manuscript. Hereafter our responses and the changes made in the manuscript according to the recommendations of the Reviewer 2 are reported. All modifications were mark with the Track Changes tool.
Response to Reviewer 2 Comments
The manuscript has been checked by a native English-speaking colleague and extensive English editing has been done.
1) Why has PEG wound healing potential similar to ChsDG on day 9?
Response 1: It is well known that PEG is water soluble and low toxic, making it suitable for contact with living organisms. In addition, PEG is widely used as plasticizer and its incorporation improve the release of poorly water-soluble bioactive substances. For this reason we have added PEG to the chitosan/diosgenin. Indeed, the PEG wound healing potential is similar to ChsDg on day 9th. Nevertheless, the ChsDg combination gave a significantly greater reduction in wound area as early as day 3 compared to PEG. In conformity with this comment, an explanation has been added in the revised manuscript.
2) There is no significant difference in the wound area of Chs and ChsDg. What could be the advantages of using both?
Response 2: In fact, the significant difference in the wound area of Chs and ChsDg is appeared on the 9th day. The advantage is that the measured mean wound area treated with ChsDg was 2.5 times smaller than this treated with Chs. This comment has been taken into consideration and additional information for mean wound area was provided in the revised version.
3) Have authors observed immune cells in wound tissues? 4) Authors should show pro and anti-inflammatory macrophages in wound tissues through immuno-staining. 5) Keratin 14/5 expression in wound tissues should be shown as these are important for skin regeneration.
Responses 3, 4, 5: The focus of the manuscript was on finding the appropriate conditions for combining the poorly water-soluble diosgenin with chitosan and studying their effect on wound healing. For that reason, we did not study in details the mechanisms of healing. Nevertheless, a more detailed effect of ChsDg combination on immune cells, macrophages and keratin intermediate filaments formation will be the topic of a forthcoming study.

Reviewer 3 Report
Major modification:
1. Please add a control group with no treatment. The 50% ethanol in the manuscript is not enough as a control, because 50% ethanol will also have an effect on wound healing.
2. In animal experiments, please redesign the grouping. It is recommended to make two wounds on the back of each mouse in the same position, and the two wounds are treated the same, and then each treatment group has at least 3 mice. Because if four wounds are made on a mouse, firstly, the wound healing in different positions is different, and secondly, if the four wounds are treated differently, there will be mutual influence between the groups.
3. It is recommended to add relevant immunofluorescence staining for samples between groups, such as neutrophils, macrophages, blood vessels, nerves, fibroblasts, collagen, etc., so that the wound healing can be better evaluated.
Minor modification:
1. In Figure 4, except for the Intact group, the picture quality of the other groups is not good, please provide a better quality picture to show the complete wound. It is recommended to increase the results of Masson's collagen staining, which can better show the condition of the wound.
2. The Discussion section should go deeper.
Author Response
POINT-TO-POINT RESPONSE TO REVIEWER 3 COMMENTS
On behalf of all the co-authors I gratefully thank the Reviewer 3 for the thorough analysis, as well as for the major and minor comments aimed at improving the quality of our manuscript. Hereafter our responses and the changes made in the manuscript according to the recommendations of the Reviewer 3 are reported. All modifications were mark with the Track Changes tool.
Response to Reviewer 3 Major Comments:
- Please add a control group with no treatment. The 50% ethanol in the manuscript is not enough as a control, because 50% ethanol will also have an effect on wound healing.
Response 1: We have determined the effect of the ChsDg on wound healing and verified that combination had a reliably significantly greater effect than that of 50% ethanol. These results indicate that probably the rate of wound closure of ChsDg will also be greater compared to untreated wounds.
- In animal experiments, please redesign the grouping. It is recommended to make two wounds on the back of each mouse in the same position, and the two wounds are treated the same, and then each treatment group has at least 3 mice. Because if four wounds are made on a mouse, firstly, the wound healing in different positions is different, and secondly, if the four wounds are treated differently, there will be mutual influence between the groups.
Response 2: Indeed, redesign of the wounds grouping on the back of each mouse in the same position will lead to the use of many more mice, which is contrary to the current trends to reduce the number of experimental animals used and was one of the requirements of our Ethics committee. However, according to the referee comment, we will evaluate this observation in future studies. In conformity with the use of a four-wound model on the back of mice, the following work could be mentioned: Masson-Meyers, D. S., Andrade, T. A. M., Caetano, G. F., Guimaraes, F. R., Leite, M. N., Leite, S. N., & Frade, M. A. C. (2020). Experimental models and methods for cutaneous wound healing assessment. International Journal of Experimental Pathology, 101(1–2), 21–37. https://doi.org/10.1111/IEP.12346
- It is recommended to add relevant immunofluorescence staining for samples between groups, such as neutrophils, macrophages, blood vessels, nerves, fibroblasts, collagen, etc., so that the wound healing can be better evaluated.
Response 3: We would like to thank the Reviewer for this remark. It will be take into consideration in future research, according to our goals and possibilities.
Response to Reviewer 3 Minor Comments:
- In Figure 4, except for the Intact group, the picture quality of the other groups is not good, please provide a better quality picture to show the complete wound. It is recommended to increase the results of Masson's collagen staining, which can better show the condition of the wound.
Response 1: This comment has been taken into consideration and Figure 4 was improved.
- The Discussion section should go deeper.
Response 2: As suggested by Referee 3 the Discussion section has been rewritten with more details.

Reviewer 4 Report
This manuscript lacks of enough quality for the publication on IJMS. Firstly, the overall design of the present study was not novel, and all the materials are widely employed. Secondly, the animal data among ChsDg, Chs, and PEG seem no significant differences (see Figures 4-7), so the addition of Dg and Chs seems nonsense and using PEG only was enough according to the current data. Thirdly, the animal model has serious flaws. Murine wounds are known to heal by contraction rather than re-epithelialization, so the murine wounds should be splinted.
Author Response
POINT-TO-POINT RESPONSE TO REVIEWER 4 COMMENTS
On behalf of all the co-authors I gratefully thank the Reviewer 4 for the valuable comments and suggestions. Hereafter our responses and the changes made in the manuscript according to the recommendations of the Reviewer 4 are reported. All modifications were mark with the Track Changes tool.
Response to Reviewer 4 Comments
This manuscript lacks of enough quality for the publication on IJMS. Firstly, the overall design of the present study was not novel, and all the materials are widely employed. Secondly, the animal data among ChsDg, Chs, and PEG seem no significant differences (see Figures 4-7), so the addition of Dg and Chs seems nonsense and using PEG only was enough according to the current data. Thirdly, the animal model has serious flaws. Murine wounds are known to heal by contraction rather than re-epithelialization, so the murine wounds should be splinted.
Response: Indeed, the used model is not new. However, there is a lack of data in the literature on the effect of chitosan/diosgenin on wound healing in experimental animals. Moreover, the results indicated that the measured mean wound area treated with ChsDg was 2.5 times smaller than this treated with Chs and PEG. Indeed, the splinted excisional wound model has been recommended as a closer to human wounds. Nevertheless, when the effect of Chs on wound healing is studied, usually did not use splint. For example, the following articles can be recommended:
Burkatovskaya, M., Tegos, G. P., Swietlik, E., Demidova, T. N., P Castano, A., & Hamblin, M. R. (2006). Use of chitosan bandage to prevent fatal infections developing from highly contaminated wounds in mice. Biomaterials, 27(22), 4157–4164. https://doi.org/10.1016/J.BIOMATERIALS.2006.03.028
Mi, F. L., Shyu, S. S., Wu, Y. B., Lee, S. T., Shyong, J. Y., & Huang, R. N. (2001). Fabrication and characterization of a sponge-like asymmetric chitosan membrane as a wound dressing. Biomaterials, 22(2), 165–173. https://doi.org/10.1016/S0142-9612(00)00167-8
Harti, A. S., Sulisetyawati, S. D., Murharyati, A., Oktariani, M., & Wijayanti, I. B. (2016). The Effectiveness of Snail Slime and Chitosan in Wound Healing. International Journal of Pharma Medicine and Biological Sciences, 5(1), 76–80. https://doi.org/10.18178/ijpmbs.5.1.76-80
Burkatovskaya, M., Castano, A. P., Demidova-Rice, T. N., Tegos, G. P., & Hamblin, M. R. (2008). Effect of chitosan acetate bandage on wound healing in infected and noninfected wounds in mice. Wound Repair and Regeneration : Official Publication of the Wound Healing Society [and] the European Tissue Repair Society, 16(3), 425. https://doi.org/10.1111/J.1524-475X.2008.00382.X

Round 2
Reviewer 3 Report
Looking forward to further improvement experiments
Author Response
POINT-TO-POINT RESPONSE TO REVIEWER 3 COMMENTS
We would like to thank Reviewer 3 for the comments and suggestions regarding the revised version of the manuscript. All modifications were mark with the Track Changes tool. Our point-by-point response is as follows:
Response to Reviewer 3 Major Comments:
- Please add a control group with no treatment. The 50% ethanol in the manuscript is not enough as a control, because 50% ethanol will also have an effect on wound healing. 2. In animal experiments, please redesign the grouping. It is recommended to make two wounds on the back of each mouse in the same position, and the two wounds are treated the same, and then each treatment group has at least 3 mice. Because if four wounds are made on a mouse, firstly, the wound healing in different positions is different, and secondly, if the four wounds are treated differently, there will be mutual influence between the groups. 3. It is recommended to add relevant immunofluorescence staining for samples between groups, such as neutrophils, macrophages, blood vessels, nerves, fibroblasts, collagen, etc., so that the wound healing can be better evaluated.
Responses 1, 2 and 3: As we already described in our initial response the redesign of the animal experiments will lead to the use of many more mice, which is in contrary to the current trends to reduce the number of experimental animals used and was one of the requirements of our Ethics committee. At this point of our manuscript processing the planning another experiment with animals would lead to significant changes in authorship and most importantly it is a time-consuming experiment making it impossible to response in time. We do not agree with the suggestion that the wound healing in different positions is different and that there will be mutual influence between the groups. We believe that the used methodology (four-wound model on the back of mice) is in accordance with those in the literature. Therefore, we ask for Reviewer’s understanding concerning our decision to take it into consideration in our future research.
Response to Reviewer 3 Minor Comments:
- In Figure 4, except for the Intact group, the picture quality of the other groups is not good, please provide a better quality picture to show the complete wound. It is recommended to increase the results of Masson's collagen staining, which can better show the condition of the wound.
Response 1: The Figure 4 was already changed. Concerning the Masson's collagen staining we completely agree with the reviewer. However, as we already pointed, the planning another experiment with animals is a time-consuming experiment making it impossible to response in time. That is why we will take it into consideration in future research, according to our goals and possibilities.
- The Discussion section should go deeper.
Response 2: This comment has been taken into consideration and additional discussion was provided in the revised version.

Reviewer 4 Report
No obvious improvement was found in the revised edition.
Author Response
POINT-TO-POINT RESPONSE TO REVIEWER 4 COMMENTS
We would like to thank Reviewer 4 for the comments and suggestions regarding the revised version of the manuscript. All modifications were mark with the Track Changes tool. Our point-by-point response is as follows:
Response to Reviewer 4 Comments
This manuscript lacks of enough quality for the publication on IJMS. Firstly, the overall design of the present study was not novel, and all the materials are widely employed. Secondly, the animal data among ChsDg, Chs, and PEG seem no significant differences (see Figures 4-7), so the addition of Dg and Chs seems nonsense and using PEG only was enough according to the current data. Thirdly, the animal model has serious flaws. Murine wounds are known to heal by contraction rather than re-epithelialization, so the murine wounds should be splinted.
Response: We believe that the research as it is presented is in the scope of the journal. In our opinion, the obtained results are enough challenging and would be of interest for the readers of International Journal of Molecular Sciences. That’s why we submitted the manuscript to the Special Issue The Emerging Role of Polymeric Materials in Pharmaceutical Designs and Applications. As we already described in our initial response the overall design of the present study is not new – it is generally accepted and applied in the research studies of new materials for wound healing. Indeed, chitosan are widely employed and will be widely employed for preparation of advanced materials because of its unique properties and functionalities. However, there is no data in the literature on the combine effect of chitosan and diosgenin on wound healing in experimental animals. Moreover, the obtained results clearly show the advantages of the prepared combination and their synergistic activity on wound healing. According to the referee second comment, additional results explanations to aforementioned Figures have been included in the revised manuscript in order to clarify the differences. Concerning the third remark, as we already described in our initial response, we agree that the splinted excisional wound model has been recommended as a closer to human wounds. However, we do not agree with the animal model has serious flaws. There are many examples in the research literature where our model (without splint) has been used specifically to investigate the effect of chitosan gels on wound healing.

Round 3
Reviewer 3 Report
Although the requirements of the ethics department are relatively strict, this is not the reason for the unreasonable design of the experiment. You can make four wounds on the mouse's back, but it is recommended to treat them all with the same drug.
Author Response
POINT-TO-POINT RESPONSE TO REVIEWER 3 COMMENTS
On behalf of the all co-authors I gratefully thank the Reviewer 3 for the valuable comments and suggestions. All modifications were mark with the Track Changes tool. Our point-by-point response to the Reviewer 3 is as follows:
Reviewer 3:
Although the requirements of the ethics department are relatively strict, this is not the reason for the unreasonable design of the experiment. You can make four wounds on the mouse's back, but it is recommended to treat them all with the same drug.
Response: As we already replied in our second response, the used methodology (four-wound model on the back of mice) is in accordance with those in the literature, e.g. Masson-Meyers, D. S., Andrade, T. A. M., Caetano, G. F., Guimaraes, F. R., Leite, M. N., Leite, S. N., & Frade, M. A. C. Experimental models and methods for cutaneous wound healing assessment. Int. J. Exp. Pathology, 2020, 101(1–2), 21-37; Sami DG, Heiba HH, Abdellatif A. Wound healing models: a systematic review of animal and non-animal models. Wound Medicine. 2019, 24(1), 8-17. Moreover, in that way we are able to eliminate the influence of the individual characteristics of experimental animals, reflected at different velocities of wound healing. Therefore, we did four wounds to the same animal to compare the process of healing after treatment with the various substances against the background of the same individual characteristics. Once again, this approach is in accordance to the current trends to reduce the number of used experimental animals and to the requirements of our Ethics committee.
We believe that the revised manuscript is in the scope of the International Journal of Molecular Sciences and in our opinion, the obtained results are enough challenging and would be of interest for the readers. That’s why we submitted the manuscript to the Special Issue The Emerging Role of Polymeric Materials in Pharmaceutical Designs and Applications.

Reviewer 4 Report
Some major concerns and issues should be further addressed before publication.
1. The authors have explained the reasons why both chitosan and diosgenin were chosen. Please also explain the reasons why PEG was chosen. In addition, why 50% ethanol was utilized as carrier.
2. As well known that, chitosan is commonly dissolved in an acidic solution. Could the 50% ethanol totally dissolve the chitosan? If acid was added, did it cause any side effects for the wound bed?
3. The photos on Day 3 and 6 are also suggested to be given in Figure 2.
4. The quality of Figure 3 should be improved, and the wound contraction ratio (%) is suggested to replace mean wound area in order to improve the readability.
5. From Figure 3-7, the results data among ChsDg, Chs, and PEG showed no significant differences, so the addition of Dg and Chs seems nonsense and using PEG only was enough according to the current data. Please justify this.
6. For animal study, the murine wounds were not splinted. This is important to mention as murine wounds are known to heal by contraction rather than re-epithelialization. These points should be mentioned and discussed in the manuscript.
7. Was the chitosan-diosgenin combination presented in the wound site as solution? How long will they stay on the wound bed?
8. The grammar and writing should be improved in the whole manuscript.
Author Response
POINT-TO-POINT RESPONSE TO REVIEWER 4 COMMENTS
On behalf of the all co-authors I gratefully thank the Reviewer 4 for the valuable comments and suggestions aimed at improving the quality of our manuscript. Hereafter the changes made in the manuscript according to the recommendations of the Reviewer 4 are reported. All modifications were mark with the Track Changes tool. Our point-by-point response is as follows:
Reviewer 4:
Some major concerns and issues should be further addressed before publication.
- The authors have explained the reasons why both chitosan and diosgenin were chosen. Please also explain the reasons why PEG was chosen. In addition, why 50% ethanol was utilized as carrier.
Response 1: In general, PEG is approved polymer from the United States Food and Drug Administration (US FDA), due to its tunable properties and well-established safety profile. Moreover, US FDA’s Inactive Ingredient Guide lists use of PEGs in oral, topical and intravenous formulations.
As we mentioned in the Introduction, PEGs are widely used as plasticizers and are suitable for contact with living organisms. One of the reasons that we have chosen PEG was in order to overcome the poor mechanical properties of chitosan. On the other hand, its incorporation in various formulations improves the release of poorly water-soluble bioactive substances (such as diosgenin) and improve the therapeutic efficacy of various medications.
Concerning 50% ethanol – it is widely used in transdermal drug delivery systems. In our experiments, it was used in order to minimize drying time in our film-forming system. On the other hand, it is a good solvent for diosgenin. That allowed preparation of viscous chitosan solution in which diosgenin (poorly water-soluble) is fully dissolved.
- As well known that, chitosan is commonly dissolved in an acidic solution. Could the 50% ethanol totally dissolve the chitosan? If acid was added, did it cause any side effects for the wound bed?
Response 2: Indeed, chitosan is commonly dissolved in an acidic solution. In our experiment, we have used lactic acid instead of commonly used acetic acid. Initially, chitosan was dispersed in distilled water followed by adding lactic acid until the complete dissolution (not in 50% ethanol). Subsequently, ethanol solutions of PEG and diosgenin were added to chitosan solution. In the final viscous chitosan solution the total water : ethanol volume ratio was 1 : 1 (or 50% ethanol). Lactic acid was used in chitosan dissolution due to its plasticizer characteristic, which gives lower stiffness and a higher percentage of elongation, besides helping the antimicrobial properties.
- The photos on Day 3 and 6 are also suggested to be given in Figure 2.
Response 3: According to the referee comment, the photos on Day 3 and 6 was added as Figure S1 as a Supplementary material.
- The quality of Figure 3 should be improved, and the wound contraction ratio (%) is suggested to replace mean wound area in order to improve the readability.
Response 4: This comment has been taken into consideration and Figure 3 was changed and improved.
- From Figure 3-7, the results data among ChsDg, Chs, and PEG showed no significant differences, so the addition of Dg and Chs seems nonsense and using PEG only was enough according to the current data. Please justify this.
Response 5: As we have written in the discussion part, it is clear that the application of ChsDg and Chs leads to a faster wound retraction (on the 3rd day), while the effect of PEG is manifested only on the 6th day. In conformity with this comment, the additional discussion was included in the revised version.
- For animal study, the murine wounds were not splinted. This is important to mention as murine wounds are known to heal by contraction rather than re-epithelialization. These points should be mentioned and discussed in the manuscript.
Response 6: Indeed, there is no animal model that represents all aspects of wound healing seen in humans. It is assumed that the main disadvantage of the excisional wound model in rodents is that the healing process is through the contraction of the panniculus carnosus while the human wound heals through re-epithelization. In this study, we applied the excisional wound model used in other publications testing the effect of chitosan (Mi et al. 2001; Burkatovskaya et al. 2006; Harti et al. 2016) without silicon splint since the splinting led to mutilation of the ring by animals’ self-grooming and the protective dressings led to high failure rate because of the motion and activity of the animals. In addition, some authors revised the considered limits of the excisional wound model because of the perception that rodent wounds heal by contraction while humans heal by re-epithelialization (Chen et al., 2015). The data have shown that contraction occurs only after epithelial closure, and the notion of the domination of closure by contraction is rather inaccurate. Thus, simple murine excisional wounds provide a valid and reproducible model that heals by both contraction and re-epithelialization (Chen et al., 2015).
This comment has been taken into consideration and the additional discussion was included in the revised version.
- Was the chitosan-diosgenin combination presented in the wound site as solution? How long will they stay on the wound bed?
Response 7: As we have written in the discussion part, chitosan-diosgenin combination was applied as a viscous solution (dynamic viscosity of 1780 cP). In fact, it was physical gel and was applied immediately after the creation of wounds. Then, before the next treatment, the wounds were cleaned with 50% ethanol and then re-applied with the same substances. This procedure was repeated at 3 and 6 (D3 and D6) days.
- The grammar and writing should be improved in the whole manuscript.
Response 8: This comment has been taken into consideration and grammar and writing were improved.
